# Mechanisms Regulating Energy Homeostasis in Plant Cells and Their Potential to Inspire Electrical Microgrids Models

**DOI:** 10.3390/biomimetics7020083

**Published:** 2022-06-19

**Authors:** Nobuhiro Suzuki, Shunsuke Shigaki, Mai Yunose, Nicholas Raditya Putrawisesa, Sho Hogaki, Maria Carmela Di Piazza

**Affiliations:** 1Department of Materials and Life Sciences, Faculty of Science and Technology, Sophia University, 7-1 Kioi-Cho, Chiyoda, Tokyo 102-8554, Japan; m-yunose-kw2@eagle.sophia.ac.jp (M.Y.); frekafreki@gmail.com (N.R.P.); s-hogaki-6b6@eagle.sophia.ac.jp (S.H.); 2Department of System Innovation, Osaka University, 1-2 Machikaneyama-Cho, Toyonaka, Osaka 560-0043, Japan; shigaki@arl.sys.es.osaka-u.ac.jp; 3Istituto di Ingegneria del Mare (INM), Consiglio Nazionale delle Ricerche (CNR), Via Ugo La Malfa 153, 90146 Palermo, Italy; mariacarmela.dipiazza@cnr.it

**Keywords:** energy, chloroplasts, mitochondria, plant cell, smart microgrid, Snf1-Related Protein Kinase (SnRK1), Target of Rapamycin (TOR)

## Abstract

In this paper, the main features of systems that are required to flexibly modulate energy states of plant cells in response to environmental fluctuations are surveyed and summarized. Plant cells possess multiple sources (chloroplasts and mitochondria) to produce energy that is consumed to drive many processes, as well as mechanisms that adequately provide energy to the processes with high priority depending on the conditions. Such energy-providing systems are tightly linked to sensors that monitor the status of the environment and inside the cell. In addition, plants possess the ability to efficiently store and transport energy both at the cell level and at a higher level. Furthermore, these systems can finely tune the various mechanisms of energy homeostasis in plant cells in response to the changes in environment, also assuring the plant survival under adverse environmental conditions. Electrical power systems are prone to the effects of environmental changes as well; furthermore, they are required to be increasingly resilient to the threats of extreme natural events caused, for example, by climate changes, outages, and/or external deliberate attacks. Starting from this consideration, similarities between energy-related processes in plant cells and electrical power grids are identified, and the potential of mechanisms regulating energy homeostasis in plant cells to inspire the definition of new models of flexible and resilient electrical power grids, particularly microgrids, is delineated. The main contribution of this review is surveying energy regulatory mechanisms in detail as a reference and helping readers to find useful information for their work in this research field.

## 1. Introduction

Energy production, storage, and management in plant cells are governed by diverse mechanisms concurring to the plant life processes in all possible environmental conditions. Flexibility of energy regulatory systems in plants relies on the ability of plants to modulate the functions of chloroplasts and mitochondria [1,2,3]. The ability of chloroplasts to absorb the light energy has a significant effect on the photosynthetic efficiency of plants [1]. Photosynthesis in the chloroplasts involves a set of reactions providing the source of usable energy for cells derived from light energy. The energy is then utilized for assimilation of carbon dioxide (CO_2_), thus generating carbohydrates as energy storage and oxygen, which supports life on Earth [4]. As sessile organisms, plants are always exposed to continuous changes in environmental conditions. Particularly, severe environments result in excess light energy in the chloroplasts, leading to the accumulation of damaging oxygen by-products. Thus, plants possess the ability to flexibly adjust light absorption and photosynthetic activities to prevent such damages [4]. For example, plant cells can accumulate a red-colored compound called anthocyanin and filter or reflect harmful radiation [5]. Chloroplasts in plant cells also possess the ability to control the activities of enzymes that scavenge damaging oxygen by-products, resulting in protection of photosynthetic activities [6]. Mitochondria are known as the main energy-producing source in all organisms. Similar to the chloroplasts, drastic environmental fluctuations can cause dysfunction of mitochondria, leading to an increase in damaging oxygen by-products [3]. Interestingly, mitochondria in plants possess specific sets of components that are not found in other organisms [3]. Such specific features of plant mitochondria might be essential for the flexible modulation of energy-producing metabolisms depending on the situations. Indeed, the unique components of plant mitochondria include enzymes that prevent accumulation of damaging oxygen by-products by modulating energy-producing metabolisms or dissipating excess electrons accumulated in the mitochondria [3,7,8].

A myriad of research has revealed that chloroplasts and mitochondria act in a coordinated manner during photosynthesis [9,10]. For example, mitochondria might also supply the energy required for carbohydrate synthesis in the chloroplasts. More generally, optimal photosynthesis requires proper balance between cellular energy and carbon [11,12,13]. Specific activities of the mitochondria were shown to be involved in the maintenance of the balance between energy, carbon, and nutrients in fluctuating environments, thereby directly optimizing photosynthesis [14,15,16]. In addition, chloroplasts also possess the ability to activate mitochondrial functions to help energy synthesis [17]. Furthermore, the mechanisms to adjust morphology and internal energy states in response to changes in environments rely on an internal capacity to sense environmental situations and to process this information [2]. Recent studies suggest that energy-producing sources, chloroplasts, and mitochondria might be tightly linked to the mechanisms that sense environmental changes [2]. It was also demonstrated that the functions of the chloroplasts and mitochondria are integrated with those of other cellular components under environmental stresses [6,18,19], suggesting the significance of energy regulatory systems in the chloroplasts and mitochondria in the adaptation of the entire cell to a fluctuating environment.

Under the current situations, in which drastic climate changes with consequent extreme natural events highly impact human societies, new types of flexible electrical power grids that are adaptive to fluctuating environments are essential for the stable supply of energy. Models of power grids resilient to fluctuating environmental conditions have been proposed in many studies [20,21,22]. In [20], the connected microgrid paradigm is proposed as a universal solution for improving the resilience of large power systems against extreme events. The analysis of different fault types in electrical power systems is presented in [21], together with the identification of some clusters of mitigation approaches related to the system resilience concept. In [22], a bio-inspired optimization model is implemented, with the aim of re-organizing electric power systems to mimic the robustness of ecosystems, namely food webs.

With the development of artificial intelligence, bio-inspired algorithms describing the concepts of biological evolution, and dynamics of energy from its sources to sinks in the biocenosis in nature or synaptic systems of the brain, have recently been getting more attention from the research communities and have shown potential for solving issues of energy production/provision [22,23,24,25,26,27]. In addition, features of innate immune response and nervous systems that provide a coordinated response to stresses were shown to be applicable for the design of smart grid models with a resiliency to unexpected dysregulation of systems [28,29]. It was also proposed that combination of feedback/forward loops of signals involved in biological homeostatic phenomena, together with autonomous regulation of coordinated movement of different muscle cells, which generate proper rhythmic patterns, is desirable for development for the complex controlling of smart grids. A new model for sustainable energy systems based on homeostatic control (HC) capabilities, which incorporates reactive and predictive homeostasis schemes, is proposed in [30]. This new model is intended to complement traditional energy control and management methods, such as droop control, to make the management system suitably equipped to meet environmental challenges and resilience capabilities. More recently, research presented an Artificial Neural Network (ANN)-based model combined with a Particle Swarm Optimization (PSO) algorithm for a Biomass Gasification Plant (BGP) that allows estimation of the amount of biomass needed to produce the required syngas to meet the energy demand [31]. The Artificial Bee Colony Algorithm has also been employed to design optimal power sharing in microgrids, resulting in minimizing the cost and power transmission from outside the microgrid with high resilience [32]. Moreover, solar cells inspired by plant cellular systems that convert light energy to the usable form have been recently proposed [33].

All these findings suggest that it is worth further exploring the possibility of using different biologically inspired models to define new modeling and energy managing approaches for electrical power grids. According to such a consideration, in this article, the mechanisms underlying energy homeostasis in a plant cell are surveyed and described in detail, highlighting peculiarities about the capability of the plant cell to produce energy, the energy consumption in plant cell, the energy storage mechanisms in plant cell, the energy regulatory mechanisms in the plant cell, and the interactions existing among plant cells. Similarities of plant cell components, mechanisms, and phenomena with structures and operations of electrical power grids, with a focus on electrical microgrids, are then put in evidence, along with the potential of the identified plant cell mechanisms to inspire the definition of electrical microgrid models accounting for resiliency and adaptivity. For the readers of this review, surveying energy regulatory mechanisms in detail should contribute to finding useful information for their work in this research field.

## 2. The Capability of the Plant Cells to Produce Energy

In plant cells, the chloroplasts and mitochondria are well-known as the main power sources (Figure 1). Photosynthesis consists of mainly two steps. In the first step, the chloroplasts can convert light energy to the usable energy and adenosine triphosphate (ATP), an organic compound that possesses a high-energy phosphate bond. In the second step, then, chemical energy storage (i.e., carbohydrates), such as soluble sugars and starches, are synthesized by utilizing ATP produced in the 1st step. Carbohydrates produced by photosynthesis are disassembled in the cytosol and the resulting product is metabolized in the mitochondria. These processes are accompanied by the production of ATP. In addition, these processes in the cytosol and mitochondria also provide electron careers that are utilized for transport of electrons through the series of proteins, activating ATP synthetic enzymes in the mitochondria [34,35]. The prioritized strategies that produce ATP can be different depending on the parts in the plant body or conditions [34,35]. ATP produced via respiration in the mitochondria, as well as its associated metabolisms, is utilized as energy to drive biological processes in various cellular components.

A plant cell possesses multiple chloroplasts and mitochondria. In a plant species, *Arabidopsis thaliana*, utilized as a model plant like a mouse for medical research, 300–450 mitochondria, and 100 chloroplasts were observed in a leaf cell [36,37,38]. Numerous studies demonstrated that the chloroplasts and mitochondria work together with other cellular components via transfer and perception of signals and energy [2,6,18,19,39,40]. These findings suggest that functions of these multiple small power sources are tightly integrated in each other or with other cellular components to finely tune the energy status in a plant cell.

Plant cells also possess alternative energy sources that generate ATP under the situations in which energy production by the chloroplasts and mitochondria is limited. For example, lipids are known as alternative sources of ATP when cells are exposed to energy depletion (Figure 1). The process to degrade lipids is called “autophagy”. In this process, lipids are surrounded by the membrane that is temporary produced during energy depletion [41,42]. The membrane including lipids is then degraded [43]. Other than autophagy, plant cells possess multiple mechanisms to target and degrade materials as needed [43,44], and products generated via degradation of target materials are utilized as energy. For example, the mechanisms to regulate amino acid degradation were shown to be triggered by signals from the mitochondria under stress conditions accompanied by energy depletion [2,41], suggesting the integration between the power sources and the alternative energy production mechanisms via “emergency signals”. In addition, plants possess mechanisms to degrade damaged mitochondria for the quality control for themselves [42]. Materials generated via mitochondrial degradations can be also recycled as new energy sources [45]. These facts suggest that plants possess multiple sources of energy production, and different sets of energy-producing mechanisms can be active depending on the situations.

## 3. The Consumption of Energy by the Plant Cells

ATP produced in plant cells is mainly utilized through the function of a type of enzyme named “kinase”. Kinases catalyzes the process known as “phosphorylation”, where a high-energy phosphate group is transferred from ATP to substrates, including proteins. Then, the substrates that perceive the phosphate group are modulated in activity, structure, localization in the cell, and stability [46]. *Arabidopsis thaliana* might possess more than 1000 kinases. Importantly, kinases and their target proteins exist in any parts of a cell and regulate a wide variety of processes [46]. In plants, two kinases, named “Target of Rapamycin (TOR)” and “Sucrose Non-Fermenting 1-Related Protein Kinases 1 (SnRK1)” are known as key regulators of growth and adaptation to environmental stresses, respectively. These kinases might modulate energy consumption depending on the energy status in cells.

TOR is known as a positive regulator of growth under normal conditions in which plant cells can produce sufficient energy (Figure 2; [47]). TOR transfers high-energy phosphate from ATP to the target proteins; then, downstream pathways of these targets to produce proteins required for plant growth are activated under the normal conditions in which photosynthesis and respiration are active, to produce sufficient energy [47,48]. Production of proteins controlled by TOR is known to be an energy-consuming processes. Thus, TOR might also function to optimize energy consumption according to the available energy sources for meeting growth demand [49,50]. For example, TOR-mediated growth of leaves utilizes both light and sugar as energy sources, although only sugar is utilized for TOR-mediated growth of roots that are hidden in the soil [51]. It should be noted that other mechanisms regulating growth of plants under optimal conditions, which link to energy metabolisms, have also been revealed in many studies [52,53]. Energy metabolism in plant cells can, therefore, be considered as a system with multiple key components, rather than TOR-centralized system.

Systems in plant cells possess the ability to flexibly adapt to a fluctuating environment. When plants are exposed to environmental stresses accompanied by energy limitation, it is necessary to switch the state of energy usage from the “normal mode” (i.e., energy-consuming growth) to the “emergency mode” (i.e., adaptation to extreme environments) [54]. Under the energy-limiting conditions, SnRK1 inhibits TOR to prevent excess energy consumption by growth and provides energy to the components of multiple mechanisms involved in the response to environmental stresses, as well as alternative ATP-producing mechanisms (Figure 2: [54,55]). SnRK1 was recently reported to regulate more than 125 proteins [56]. Indeed, these proteins include kinases and ones previously evidenced to be activated by kinases. In addition, energy distribution in a plant cell could be also strictly modulated depending on the types of stresses, because localization of SnRK1 in different parts of a cell under different conditions has also been reported in several studies [57,58].

These results indicate that ATP produced from multiple sources is used for growth under normal conditions and responses to environmental stresses. In addition, energy usage in the regulation of growth and stress responses is strictly balanced by the functions of TOR and SnRK1.

## 4. Energy Storage in the Plant Cells

In plant cells, energy can be stored as soluble sugars, starches, and lipids. Particularly, starch, a long chain composed of glucose, is considered as main long-term energy storage in plants, with no chemical or osmotic disturbance to the cell due to water insolubility [59,60,61]. Indeed, the harvested parts of the crops such as beans and grains are starch-storing organs (seeds), and starch is one of the main contributors to the human diet in terms of calories. Starch can be also synthesized in the chloroplasts of leaves and stored as granules [60]. Shape and number of starch granules in a chloroplast can be different depending on ages of leaves and types of tissues. In addition, lack of enzymes involved in the starch granule synthesis can result in less numbers, but larger sizes, of granules in a chloroplast [60]. These findings suggest that starch might be stored with a suitable form depending on the situations.

In leaves, starch degradation occurs in the chloroplasts to produce maltose, a unit of two glucose molecules. Then, maltose is transported to the cytosol and further metabolized to other sugars such as glucose and sucrose [62]. Although packed in water-insoluble starch granules, glucose molecules can be easily accessed as needed to prevent starvation via the function of starch-degrading enzymes [63]. Glucose serves as not only a building block for starches, but also short-term energy storage that is directly associated with the processes to produce ATP via the functions of the mitochondria [59]. Glucose is also known to be the main component of the cell wall and the starting material for the synthesis of amino acids [59].

Although glucose is an important form of a carbohydrate that is actively consumed or stored in the cells, carbohydrates are transported mainly as sucrose, intracellularly or between cells. Sucrose is a favorable form for transport because it can contain more energy compared to glucose. Thus, sucrose is more energy efficient for transport and storage. In addition, sucrose does not interfere with the function of other molecules, in contrast to glucose that can cause a non-enzymatic reaction [59], leading to the alteration in the functions of important proteins [64,65,66]. 

Many plants accumulate storage lipids, usually in the form of triacylglycerols, that are often regarded as the primary source for the huge diversity in fatty acids [67]. As mentioned above, lipids are utilized as an alternative source of ATP (See the section above “2. The capability of the plant cells to produce energy”). It is also well known that triacylglycerols stored in seeds are remobilized to fuel small plants of the next generation after the germination [68]. Immediately after the germination, and prior to initiation of photosynthesis, growth of the small plants relies exclusively on triacylglycerols in seeds, which are utilized as a source of carbon skeletons and are an energy source [69,70].

These results indicate that plants possess ability to alter the form of energy storage depending on the purposes.

## 5. Energy Regulatory Mechanisms in Plant Cells

In plant cells, energy production and consumption can be optimized depending on the situations or types of cells by sophisticated mechanisms. For example, communications of chloroplasts or mitochondria with the nucleus, the control center of the cell, have been extensively studied. Functions of the chloroplasts and mitochondria are mainly regulated by the nucleus [71,72]. However, it has been well known that signals from the chloroplasts and mitochondria that regulate nuclear functions also exist [15,73,74]. Such bi-directional signals are essential to prevent production of excess energy that can damage chloroplasts and mitochondria under environmental fluctuations [15,73,74]. In addition, it has been demonstrated that energy consumption is modulated by feedback mechanisms. A recent study revealed the negative feedback systems in which a protein that is activated by TOR inhibits TOR [75]. It might be an important system to prevent excess energy consumption and growth. By the mechanisms involving these feedback systems, activities of multiple pathways that simultaneously function and, in some cases, interact with each other, are strictly balanced to finely tune energy production, distribution, and usage depending on the requirement of energy for growth and responses to environmental stresses (Figure 2; [54,55,57,58,73,74]).

TOR was shown to selectively regulate diverse branches of growth mechanisms depending on upstream signals associated with endogenous (cellular) and exogenous (environmental) energy status [48]. For example, TOR-dependent mechanisms that promote root growth are positively regulated by sugars in the cell [76]. In contrast, certain proteins in leaves were shown to be negatively regulated by sucrose in the cell, but positively regulated by light-dependent signals in TOR-dependent manner [77,78]. These findings suggest that TOR might properly activate prioritized mechanisms underlying growth, depending on the energy status in the cell and environment, as well as types of cells. Numerous studies demonstrated the significance of several compounds, called plant hormones, in the regulation of plant growth via the mechanisms regulated by TOR [47]. Among these, a plant hormone named “auxin” was shown to positively regulate TOR protein under sufficient light conditions [78]. In addition, trehalose-6-phosphate (T6P), an intermediate of sugar (trehalose) synthesis, is also shown to positively regulate TOR protein [48]. Auxin and trehalose play essential roles to regulate plant growth and development [79,80,81]. Interestingly, functions of auxin and T6P are associated with sensors of light and sugars, respectively [47]. Thus, plants possess two different types of sensors that monitor the energy status of both environment and the cell. By balancing the functions of these sensors, the activity of TOR might be finely tuned.

As mentioned above, the status of energy is switched from normal mode to emergency mode by the functions of SnRK1. Operations downstream of the SnRK1 are then tailored depending on the types of environmental stresses after switching the mode. Abscisic acid (ABA), a plant hormone, is known as a key regulator of responses of plants to stresses such as drought, salinity, extreme temperature, and so on [82]. These stresses cause damages on the chloroplasts and mitochondria, leading to a disruption of energy production in the cells. ABA functions together with SnRK1 and regulates downstream pathways to protect cells against stresses, or to allow cells to adapt to environmental stresses accompanied by energy depletion [56]. Importantly, ABA, together with several kinases, is known to regulate a wide variety of signals involved in responses of plants to environmental fluctuations [83,84,85]. In these mechanisms, different sets of kinases and their target proteins are activated depending on types of stresses [86,87,88]. ABA might play key roles to tailor the proper mechanisms in which certain pathways are prioritized, depending on the types of stresses [83,84,85]. 

Many researchers have comprehensively analyzed alterations of gene expression in plants, in response to various environmental stresses, including extreme temperatures, excess light, and so on [89,90,91]. By using these data sets, we can generate the network of genes that function in different cellular components with different roles. Figure 3 shows the networks of genes associated with the production/usage of energy and carbohydrate (i.e., energy storage) metabolisms in plants, whose activities were up-regulated by low temperature or excess light in Arabidopsis thaliana [92]. These networks demonstrate that different components (cellular components) can exchange signals or energy via the functions of devices (proteins) to constitute the systems that modulate energy status in a whole cell. Genes that are involved in the regulation of the functions of mitochondria, chloroplasts, and cytosol are mainly up-regulated in response to a low temperature in the network of production/usage of energy. Particularly, genes in cytosol and mitochondria are linked to many genes, suggesting that mitochondria and cytosol might function to modulate energy homeostasis by regulating mechanisms that function in other cellular parts. Interestingly, it was proposed that mechanisms to sense low temperature could exist in the mitochondria to modulate energy metabolisms in the cells [93]. In addition, a molecular network of carbohydrate metabolisms under a low temperature also shows the importance of genes that function in cytosol and chloroplasts. In response to excess light, genes that are involved in the regulation of the functions of the chloroplast and cytosol are mainly up-regulated. Especially, chloroplasts might play key roles in the regulation of other genes that function in other cellular components, both in the network of production/usage of energy and carbohydrate metabolisms. This tendency is reasonable, because excess light can damage chloroplasts, and mechanisms to protect photosynthetic machineries can be activated. In contrast to the mechanisms up-regulated in response to low temperature, genes involved in the mitochondrial functions do not seem to play key roles. These results support the fact that plants possess the ability to flexibly adjust energy regulatory systems depending on the types of environmental stresses. Furthermore, activity of genes involved in energy regulation is known to be flexibly altered depending on the durations of environmental stresses (Figure 4). Taken together, these facts suggest that plants are able to flexibly adapt to environmental fluctuations by modulating temporal and spatial coordination between multiple components involved in the energy regulatory mechanisms.

## 6. Interactions among the Plant Cells

In plants, transport of materials through multiple cells is necessary for the maintenance of energy status at the whole plant level. Carbohydrates produced in leaves are transported mainly as sucrose by three different ways: (1) sucrose transporters, (2) channel-connecting cells named “plasmodesmata”, or (3) vascular bundles (Figure 5; [59,94]). Sucrose synthesized in leaf cells is transported to other cells, and then leaches to the vascular bundles through the plasmodesmata [95]. In addition, sucrose is also transported through multiple cells via sucrose transporters located on the cell membrane, and it leaches to the cells adjacent to vascular bundles [95]. Sucrose loaded to the vascular bundles is then transported to other leaves, roots, and seeds. Different sucrose transporters exist in different types of cells that constitute different parts of a plant [94]. In addition, activity of these transporters is flexibly altered depending on the environmental conditions [61]. These findings suggest that carbohydrates synthesized via photosynthesis in the chloroplasts of leaves can be transported to the parts which require energy, and their distributions can be orchestrated at the whole plant level [59,61,94].

Plants possess the ability to discard parts that are no longer necessary, such as old leaves during autumn and severely damaged leaves. If a leaf is damaged or becomes too old, a plant discards it to conserve water and photosynthetic efficiency, depending on the energy cost to the whole plant body. Abscission of leaves involves regulation of the transport of a growth hormone auxin through multiple cells. When a leaf is young or healthy enough with a high activity of photosynthesis, auxin is transported from this leaf to its basal part [96]. However, in old or damaged leaves, this auxin transport stops and a layer of cells with weak cell walls is formed at the base of the leaf [96,97]. 

When a small group of cells is exposed to environmental stimuli, long-distance signals propagate through a whole plant body [98]. Then, mechanisms of stress response are activated in the parts of plant that are not directly exposed to environmental stimuli. Propagation of long-distance signals was shown to be accompanied by energy-generating and -consuming mechanisms. For example, the electric signal is known as one type of long-distance signals activated by stress. It might be associated with the transport of electrons from inside to outside the cells, which generates electric potential along the path of long-distance signaling [98,99]. In addition, kinases required for the transfer of signals from certain cells to their neighboring cells are found [98]. Long-distance signaling in plants is a flexible system without one central authority. Any parts of a plant body which are subjected to stimuli can be the starting point of the long-distance signaling. In other words, it should be a system in which a master regulator and hierarchy can be flexibly altered depending on the situations.

## 7. Similarities between Plant Cells and Microgrids

In this section, the smart microgrid concept is firstly described in brief, pointing out the difference of this paradigm with respect to conventional power grids. Furthermore, the main components and requirements of a microgrid are introduced. Similarities between energy-related components/processes of plant cells and microgrids are then put in evidence with the aim of evaluating the potential of homeostatic cell phenomena to inspire future bio-inspired modeling approaches for resilient, reliable, and efficient microgrids.

### 7.1. Main Features of Smart Microgrids

Conventional power grids are generally used to deliver the electrical power produced by a limited number of centralized high-power generation plants to many users. In these grids, the use of Information Communication Technology (ICT) and the penetration of distribution automation is limited to a small percentage (15–20%) and the data flow, as well as the power flow, is unidirectional [100]. Unfortunately, conventional power grids are not resilient to system anomalies or extreme environmental events; particularly, failures of electric power supply and, in the worst-case scenario, blackouts can occur in a broad range of areas due to domino effect failures. The smart microgrid paradigm has gained interest in the last decade as a promising solution for a progressive decarbonization of the energy mix and a more efficient, flexible, and economic operation of electrical power systems. Smart microgrids also have potential to solve problems of conventional power grids by the standpoint of the system resilience. 

A microgrid can be defined as a small-scale self-controlled power system interconnecting generators and loads within defined electrical boundaries. Microgrids interact with the upstream main grid through points of common coupling and can be operated either in grid-connected mode in coordination with the external power system or in islanded mode functioning independently. In contrast to conventional power grids, smart microgrids include multiple distributed power sources, comprising renewable generators and energy storage systems, and are equipped with a communication infrastructure, allowing to the microgrid’s components exchanging information with each other in a bi-directional manner. This allows, for example, final users of electrical energy to be involved in the electrical market and, more generally, it provides the possibility of modulating electrical power flows (according to defined optimization objectives) by suitable energy management systems (EMSs). For this to be put into practice, smart microgrids make extensive use of information and communication technology (ICT). A crucial component of a smart microgrid is the energy storage system (ESS), the function of which is to support the microgrid under several operating scenarios, e.g., by smoothing out the intermittent generation of renewable sources, by providing voltage support and frequency regulation to ensure grid stability, by allowing for end-user active participation in the energy market, and by contributing to the grid resiliency if used as a backup supply system. Important features of smart microgrids include a set of energy infrastructures and ICT technologies that support (1) generation, supply, and intelligent use of electricity; (2) metering, monitoring, and advanced management of electricity; and (3) advanced communication systems. In addition, smart microgrids are also characterized by a smart protection system, the subsystem that provides advanced grid reliability analysis, fault protection, and privacy protection services [101]. Figure 6 gives a schematic view of the main differences between conventional power grids and smart microgrids.

By the point of view of control and management, microgrids are usually governed according to a multi-level approach covering different technical areas, time scales, and physical levels. This control scheme, called hierarchical control, is composed of three main control layers, i.e., primary control, secondary control, and tertiary control, allowing for the achievement of the most significant microgrid objectives, such as voltage/frequency regulation, power sharing, synchronization, and resilient and profitable operation [102]. Hierarchical control is achieved by simultaneously using the local control of the electronic power interfaces between the microgrid and its main components (generators, storage systems, and loads) and the coordinated control of the different internal and external components of the microgrid. Secondary and tertiary control levels, in particular, exploit the coordinated interaction of some or all local controllers, thus requiring a fast and reliable communication system. According to their different modes of communication, the coordination can also be divided into three categories: centralized, decentralized and distributed [102]. It is necessary to take into account the possibility of operation in island mode and, therefore provide a logic control that can disconnect the microgrid from the rest of the system. However, the smart grid needs to be re-connected to the main grid after the restoration [103]. Figure 7 illustrates a scheme of the microgrid hierarchical control, including specific functions within each hierarchical layer.

### 7.2. From Plant Cell to Microgrid

Based on the excursus of the mechanisms underlying production, use, and energy management in plant cells, it is possible to draw out several elements of similarity with the smart microgrid paradigm. These similarities are shortly outlined hereinafter. 

### 7.3. Energy Production

Energy producing systems involving many small power sources in plant cells are analogous to the scheme of distributed energy generation in microgrids, whose advantageous operation can be achieved by implementing suitable energy management strategies, implying a coordinated use or dispatch of available energy resources.

Plant cellular systems employing auxin and T6P that monitor external and internal energy status, respectively, can be related to microgrids, including renewable sources such as photovoltaic and wind generators. In a smart microgrid encompassing distributed renewable generators, it is important to monitor meteoclimatic variables related to electrical power production from renewables such as solar radiation and wind speed, as well as electrical power demand from the load. The former could be associated with environmental conditions and the latter with internal conditions. 

### 7.4. Energy Management and Consumption under Different Environmental/Internal Conditions

As explained in Section 3, TOR might properly activate prioritized mechanisms underlying growth, depending on the energy status in the cell and environment. Thus, further in-depth study and mimicking of the mechanisms regulated by TOR could lead to the establishment of systems to regulate intelligent usage of energy in a smart microgrid. Here, we introduce a TOR-dependent mechanism as an example of energy-dependent growth regulation because systems that switch operations depending on energy states can be related to smart microgrids’ operation. All these mechanisms could be related to the operation of a microgrid under the supervisory control of an EMS, performing the coordination of demand and supply, with the aim of fulfilling economic/environmental objectives, under favorable conditions. In addition, the mechanisms regulated by TOR and SnRK1, as well as ABA, are key for the flexible mode of energy regulatory systems in plant cells, which are able to respond properly even to adverse conditions. In this respect, a similarity with the microgrid could be found in the operation of a supervisory control preserving safety, reliability, and survivability of the microgrid, which are associated to a microgrid design contemplating redundancy and backup solutions to guarantee resiliency of the system under any operating condition, including adverse extreme events. It is worth noting that both centralized and distributed architectures for energy management have been successfully used in microgrids and can be considered as candidate case studies [104]. Further elucidation of the mechanisms regulated by TOR and SnRK1 is required for the establishment of strategies to generate plant-inspired smart microgrid models or management strategies.

### 7.5. Energy Storage

Energy storage systems that are crucial for growth and survivability are observed in plant cells; analogously, smart microgrids need efficient storage of energy for their operation. In plants, lipids are essential as energy storage as well as components of cellular membranes and signaling molecules [43]. Although it is challenging to establish large-scale ESSs, mainly based on batteries, the development of new smart microgrids and storage battery systems should be proceeded in parallel. As a matter of fact, the use of battery storage systems as backup generators confer resiliency and robustness to the grid infrastructure, which are also pivotal features in a scenario of extreme events, tied to climate change. This issue will also imply economic considerations related to the microgrid design and establishment costs, even if a very high value is usually related with improved grid resiliency, especially when ensuring that supply to critical loads can prevent the loss of life [105]. Similarly, microgrid design including redundancy criteria could benefit from the backup mechanisms observed in plant cells.

### 7.6. Communication Issues and Multi-Microgrid Power Systems

When a single plant cell is regarded as a community (such as a town, a city or so on), the existence of long-distance linking signals among cells can be assumed. When an energy outage occurs in a community, the information of energy status is transferred to other communities. Then, energy status in the community in which the power outage occurs and its surroundings can be optimized. To establish such a system, feedback signals from distal parts to a local part that is directly exposed to stimuli should be elucidated in the field of plant biology. However, such feedback signals have not been revealed in hitherto studies, and elucidation of these signals is urgent for plant biologists. Even if parts of energy transportations are disrupted, each cell, tissue, or organ is able to manage energy survival by modulating their own metabolites. In many cases, these disruptions in transportation can negatively affect growth, but not lethally. Alternative mechanisms (unknown in detail) may complement these disruptions. Although the energy regulatory systems that function in a single cell could be a basis for describing analogous mechanisms in smart microgrids, features of multicellular systems that function in a whole plant body could also be a key to generate a plant-inspired model of multi-microgrid power systems.

Furthermore, the mechanism related to cell-to-cell communications and long-distance signaling can be related to the high-level coordinated operation of microgrids in larger electrical power systems which were conceived according to the smart grid paradigm. This consideration could be applied to the studies focusing on the previously introduced hierarchical control. Mechanisms regulating long-distance signals in plants could then provide an inspiring paradigm for designing the innovative hierarchical control for flexible and resilient multi-microgrid power systems.

It should be observed that all the described analogies between such different systems have the scope to catch the possibilities offered by the plant cell energy mechanisms to be exploited for defining behavioral models or control/management strategies for electrical microgrids. The research on this topic is currently at a very early stage and much work needs to be done to suitably formalize the models of energy homeostasis in plants for applying them to the microgrid field.

## 8. Conclusions

The ability of plants to acclimate to and survive under fluctuating environments relies on flexible energy regulatory systems involving multiple sources of production with backup systems; a regulatory hub that switches the energy status to activate prioritized mechanisms depending on conditions; different forms of energy storage and cell-to-cell communication systems to transport energy storage; and signals which orchestrate energy distributions in the whole plant level. Based on these characteristics described in this paper, the new idea to use the mechanisms underlying energy homeostasis in plant cells for the establishment of models designing a reliable, efficient, and resilient smart grid is proposed. For example, the mechanisms to activate prioritized pathways are tightly linked to the systems to monitor environmental and internal energy status. These systems can be related to a smart microgrid employing distributed renewable generators with monitoring systems of meteoclimatic variables and electrical power demand from the load. In addition, such plant cellular mechanisms could lead to the establishment of systems to regulate intelligent usage of energy in a smart microgrid. Furthermore, cell-to-cell communications and long-distance signaling could be a key to generate multi-microgrid power systems. These analogies between such different systems provide us hints to catch the possibilities offered by the energy regulatory mechanisms in plant cells to be exploited for defining new models of electrical microgrids. The research on this topic is currently at a very early stage and much further work is required to suitably formalize the models of energy homeostasis in plants, for applying them to the microgrid field. However, surveying energy regulatory mechanisms in this review provides useful information for readers to inspire new models of microgrids for future research.

We can expect that mimicking plant cellular systems could be more advantageous for designing new smart microgrid models when compared with animal cellular systems, since, in plant cells, mechanisms for adapting to the fluctuating environments could be more flexible compared to animals because of their sessile lifestyle [106]. Even if parts of energy regulatory systems are disrupted, plants are able to manage energy to survive via the complementation of alternative mechanisms (unknown in detail). Such an ability to survive relies on flexible molecular networks and modulation of them depending on timing. The data for the activity of genes involved in the molecular networks in plants that are exposed to environmental stresses for different durations can be obtained from the databases (Figure 4). In addition, relationships among genes or possible cellular components in the molecular networks involved in the energy regulatory systems can be also obtained (Figure 3). By combining time course data of gene activity with the molecular networks, we may be able to generate the models expressing the behavior of plant energy metabolisms under fluctuating environments. Then, we can attempt to apply these models to establish new smart microgrids. For such modeling, it is necessary to generate algorithms expressing the structure of molecular networks and time-dependent gene activities. In a previous study, bio-inspired modeling of social networks based on Particle Swarm Optimization (PSO), Ant Colony Optimization (ACO), Artificial Bee Colony Optimization (ABC), and the Firefly Algorithm (FA) have been already established [84,107]. It should be interesting to test if a similar approach is applicable for the modeling of energy metabolisms in plant cells. Furthermore, it is also necessary to identify, as rigorously as possible, all possible analogies and relationships allowing description of smart microgrid operation according to the plant-cell-modeled phenomena.

Although this article suggests new ideas for modeling smart microgrids, it is a suggestion from a plant biologist. It is indisputable that efficient collaboration between specialists of smart grid and plant biology is necessary. The purpose of this review is provision of the underpinning ideas of this new concept, not the discussion of detailed mechanisms that function in plant cells. For the detail of mechanisms regulating plant energy homeostasis, more comprehensive review articles [2,39,40,54,58,108] are recommended.

## Figures and Tables

**Figure 1 biomimetics-07-00083-f001:**
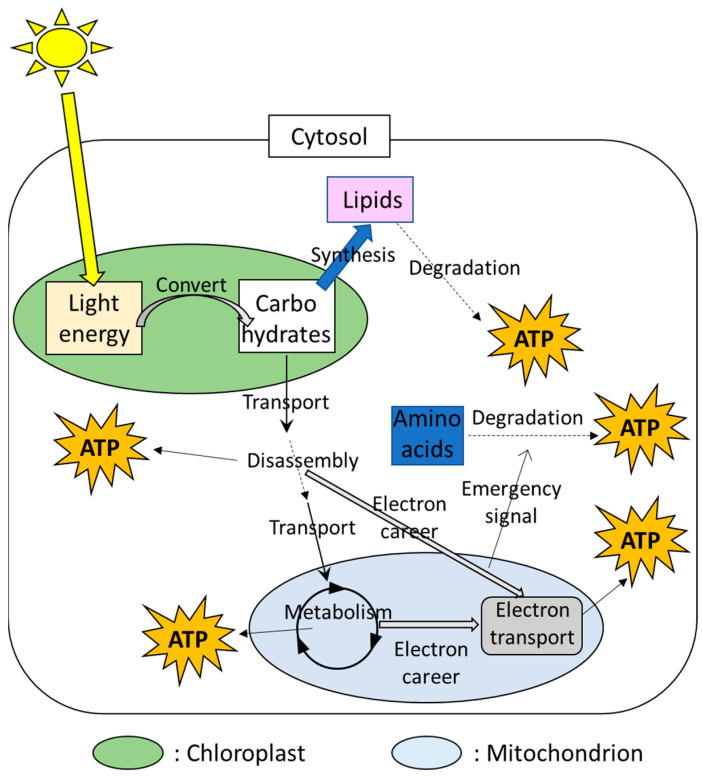
Multiple ATP-producing systems in a plant cell involving the chloroplasts and mitochondria.

**Figure 2 biomimetics-07-00083-f002:**
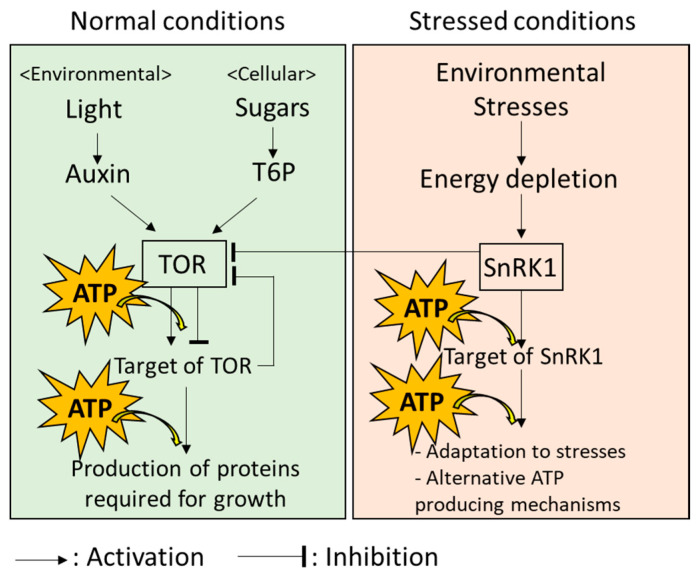
Energy-modulation systems in plant cells involving TOR and SnRK1.

**Figure 3 biomimetics-07-00083-f003:**
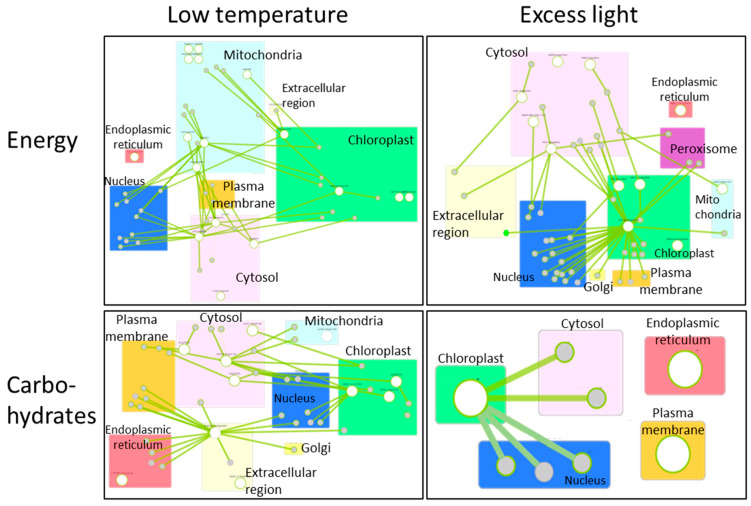
Molecular networks activated in plant cells, in response to low temperature (**left**) or excess light (**right**). Lists of genes up-regulated in response to low temperatures or excess light were obtained from a previous study in which alterations in the activities of genes were comprehensively analyzed in plants subjected to low temperature or excess light [92]. Genes involved in production/usage of energy (upper panels) and carbohydrate metabolisms (lower panels) were then picked up from the lists of low temperature or excess light responsive genes based on the information from the database, Gene Ontology Resources (http://geneontology.org/ accessed on 15 January 2021). These genes were then analyzed in the network using the database, Arabidopsis Interaction Viewer 2.0 (http://bar.utoronto.ca/interactions2/ accessed on 15 January 2021). In this database, links between genes of interest can be automatically detected based on the information from hitherto studies, and figures indicating gene networks can be generated. Large white circles indicate genes in the list of low temperature or excess light responsive genes [92] analyzed by the database. Small gray circles indicate genes that were not in the list but detected as genes linked to large white circle genes. Rectangles with different colors indicate different cellular components. Green lines indicate the links between genes.

**Figure 4 biomimetics-07-00083-f004:**
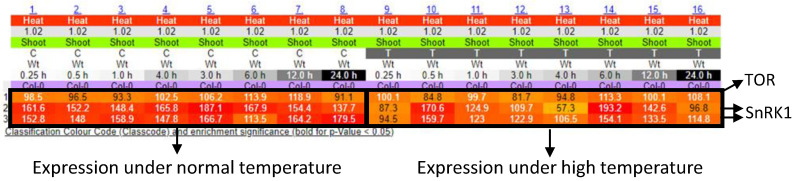
Time course expression patterns of SnRK1 and TOR genes in response to high temperature. The results were obtained from the e-Northerns w. Expression Browser (http://bar.utoronto.ca/affydb/cgi-bin/affy_db_exprss_browser_in.cgi accessed on 20 January 2021).

**Figure 5 biomimetics-07-00083-f005:**
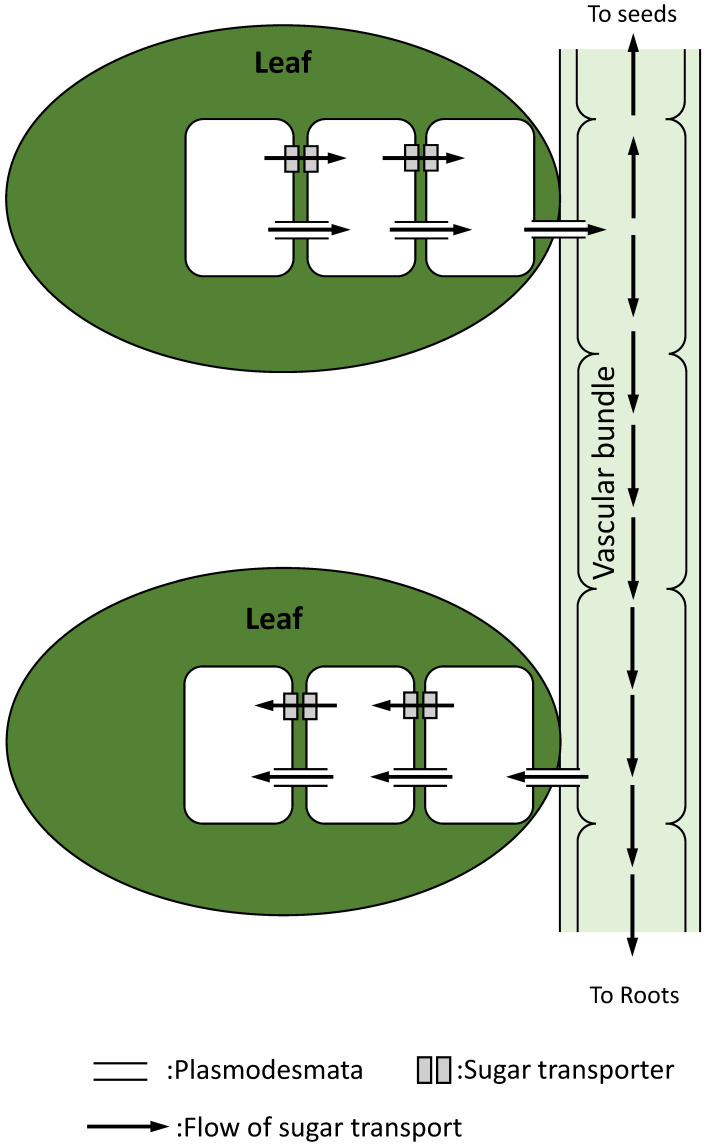
Sugar transport via cell-to-cell communications. Sucrose synthesized in a leaf can be transported through multiple cells via plasmodesmata or transporters, and it leaches to a vascular bundle.

**Figure 6 biomimetics-07-00083-f006:**
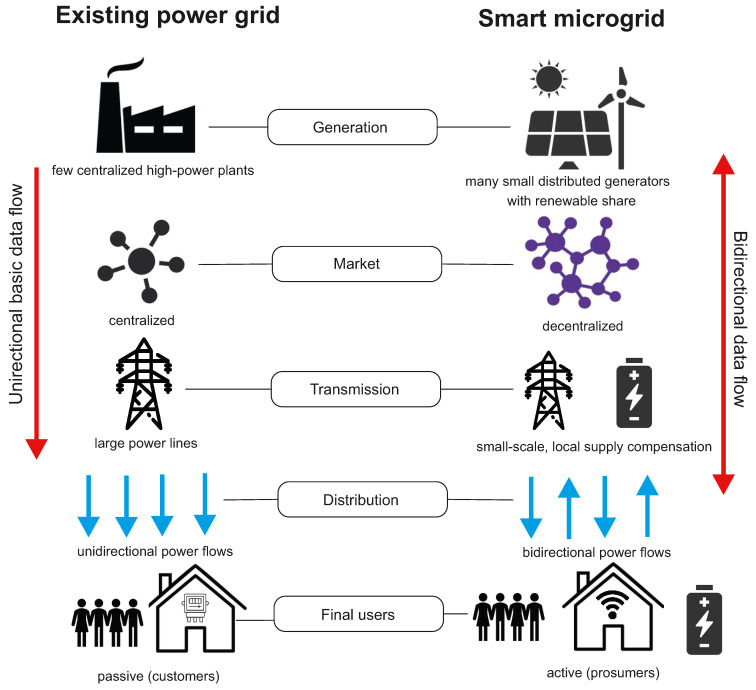
Existing power grid and future smart microgrid.

**Figure 7 biomimetics-07-00083-f007:**
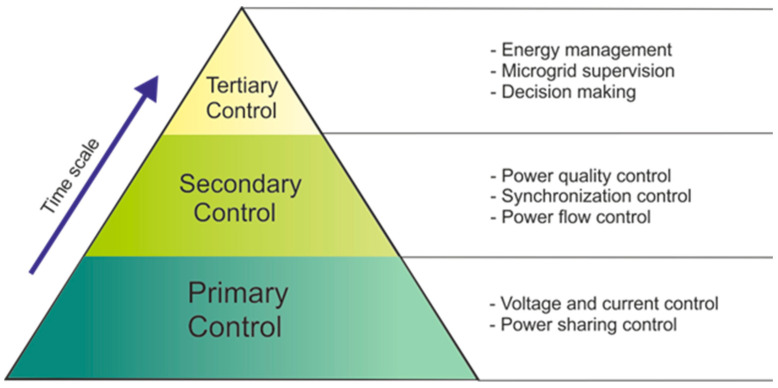
Scheme of the hierarchical control in a microgrid.

## Data Availability

Not applicable.

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
