# Peer review of "Mechanisms Regulating Energy Homeostasis in Plant Cells and Their Potential to Inspire Electrical Microgrids Models"

_biomimetics, 2022, doi:10.3390/biomimetics7020083_

Round 1

Reviewer 1 Report

The authors provided a typescript on the subject Mechanisms regulating energy homeostasis in plant cells and 2 their potential to inspire electrical microgrids models. Authors the presented main features of systems that are required to flexibly modulate energy states of plant cells in response to environmental fluctuations are surveyed and summarized. I positively assess the theses adopted by the Authors. The authors made an in-depth review of contemporary literature. The included literature references supplement the observations and research goals. I recommend that you review Figure 3 again. Some of the content presented in it is illegible.

Author Response

The authors provided a typescript on the subject Mechanisms regulating energy homeostasis in plant cells and 2 their potential to inspire electrical microgrids models. Authors the presented main features of systems that are required to flexibly modulate energy states of plant cells in response to environmental fluctuations are surveyed and summarized. I positively assess the theses adopted by the Authors. The authors made an in-depth review of contemporary literature. The included literature references supplement the observations and research goals.

The authors would like to thank the Reviewer for the performed review and for the appreciation

I recommend that you review Figure 3 again. Some of the content presented in it is illegible.

The authors would like to thank the Reviewer for this suggestion. Figure 3 has been suitably modified to make its content clearer and fully readable.

Reviewer 2 Report

A bio-inspired model is of importance to bring new solutions to many engineering fields while the system theory serves as an underlying basis. The paper has potential value for the power industry by comparing plant cells with microgrids. The following issues are recommended to be investigated further.

  1. Mapping schemes to show the similarity between the microgrid and plant cell, and that between smart grid with microgrid and multiple plant cells are desired to be provided.
  2. Certain case studies are expected to be provided to demonstrate the effectiveness of the plant cell-inspired model for the smart grid under different conditions.
  3. It is not clear how the detailed energy regulation methods in the plant cells are used for energy management of microgrids. 

Author Response

A bio-inspired model is of importance to bring new solutions to many engineering fields while the system theory serves as an underlying basis. The paper has potential value for the power industry by comparing plant cells with microgrids. The following issues are recommended to be investigated further.

The authors are grateful to the Reviewer for time spent in reviewing the manuscript and for the appreciation on the manuscript’s topic.

Mapping schemes to show the similarity between the microgrid and plant cell, and that between smart grid with microgrid and multiple plant cells are desired to be provided.

Thanks for your valuable suggestion. However, the main purpose of this review is to survey the mechanisms underlying flexible energy metabolisms in plants to provide information for readers to inspire new models of microgrids for future researches, not giving new models or results.

Certain case studies are expected to be provided to demonstrate the effectiveness of the plant cell-inspired model for the smart grid under different conditions.

The research on this topic is currently at a very early stage and case studies are still not available. This review provides information that might be useful to propose new models.

It is not clear how the detailed energy regulation methods in the plant cells are used for energy management of microgrids.

We suggested future approaches for this research. However, much further work is required to formalize the models of energy homeostasis in plant cells for the application to the microgrid designing.

Reviewer 3 Report

Dear Authors.

This paper is good Review paper (Suggestion from plant biologist: plant-inspired models of smart microgrid). But, I decision reconsider after major revision.

Strength of this paper included:

The strength of the paper included: the topic is good and interesting.

Weakness of this paper:

#1. Introduction

The "Introduction" information presented is not new.

I recommend additional/rewrite "Introduction".

-Need nice story of Introduction.

#2. Review-related statistical programs and reasons for selecting review articles

What kind of statistical programs did you use?

-Please add/more version and description.

#3. Abstract and contribution: Poor Abstract.

I recommend additional/rewrite "Abstract and contribution".

-Contribution need supported by data and result.

-Need nice story of Contribution.

#4. Methods.

-They should be described with sufficient detail to allow others to replicate and build on published results. New methods and protocols should be described in detail while well-established methods can be briefly described and appropriately cited. Give the name and version of any software used and make clear whether computer code used is available. Include any pre-registration codes (Methods).

#5. Related work: Improve (Study more)

-Important aspect has been mentioned.

-Also, more to 15 new papers (Journal) published from 2020~2022 by major publishers such as MDPI, Elsevier, Springer, IEEE, ACM, and Wiley.

    -More New Journal (2021~2022) papers.

6. Results

Results need clearly.

-Need nice story of results.

7. Other

Scientific Soundness: Low.

8. English

-English language and style are fine/minor spell check required.

9. Conclusion

-You need to write more of the conclusion part. 

-Future work" write more. (Must be improved) 

-Conclusions supported by the results. (Write more)

Author Response

Dear Authors.
This paper is good Review paper (Suggestion from plant biologist: plant-inspired models of smart microgrid). But, I decision reconsider after major revision.

Strength of this paper included:
The strength of the paper included: the topic is good and interesting.

The authors are grateful to the Reviewer for time spent in reviewing the manuscript and for the appreciation on the manuscript’s topic.

Weakness of this paper:
#1. Introduction
The "Introduction" information presented is not new.
I recommend additional/rewrite "Introduction".
-Need nice story of Introduction.

Thanks for your valuable observation. The authors have revised the manuscript introduction, adding a more detailed description of the paper scope and additional recent/relevant references.

#2. Review-related statistical programs and reasons for selecting review articles
What kind of statistical programs did you use?
-Please add/more version and description.

We did not use any statistical programs. This article is not a research paper in which new results are presented.

#3. Abstract and contribution: Poor Abstract.
I recommend additional/rewrite "Abstract and contribution".
-Contribution need supported by data and result.
-Need nice story of Contribution.

Thanks for your valuable observation. The authors agree that a clear statement of the manuscript scope/contribution is pivotal. On such a basis, a more extensive description of the scope of the proposed review is provided in the revised version of the manuscript. On the other hand, this paper is conceived as a review on energy production, storage and managing mechanisms of the plant cell with some notion in the end on how it can potentially inspire research on microgrids. Therefore, as a review paper, its main contribution is surveying in detail the energy mechanisms under consideration and help readers to find useful references for their work in this field; as a review paper this manuscript do not have the ambition to produce and show novel results itself.

#4. Methods.
-They should be described with sufficient detail to allow others to replicate and build on published results. New methods and protocols should be described in detail while well-established methods can be briefly described and appropriately cited. Give the name and version of any software used and make clear whether computer code used is available. Include any pre-registration codes (Methods).

Thanks for useful remarks. In our case, a review paper has been submitted, where a survey of mechanisms regulating plant cells homeostasis is basically developed. Therefore, no novel/original simulations or software based protocols have been specifically created and presented.

#5. Related work: Improve (Study more)

Thanks for your observation. The proposed point of view is related to the possibility to inspire studies on microgrids based on mechanisms regulating plant cells homeostasis; this is an almost unexplored topic and very few closely related works are available in technical literature.

-Important aspect has been mentioned.

Thanks for your appreciation

-Also, more to 15 new papers (Journal) published from 2020~2022 by major publishers such as MDPI, Elsevier, Springer, IEEE, ACM, and Wiley.

    -More New Journal (2021~2022) papers.

Thanks for your suggestion. We have updated the reference list by adding 13 new recent (2020-2022) and relevant papers responding to the suggested criteria. In particular, the following references have been added:

Yafei Shi, Xiangsheng Ke, Xiaoxia Yang, Yuhan Liu, Xin Hou (2022) Plants response to light stress. J Genet Genomics, 2022, S1673-8527(22)00137-0.

Stefania Fortunato, Cecilia Lasorella, Luca Tadini, Nicolaj Jeran, Federico Vita, Paolo Pesaresi, Maria Concetta de Pinto (2022) GUN1 involvement in the redox changes occurring during biogenic retrograde signaling, Plant Sci, 320:111265.

Glenda Guek Khim Oh, Brendan M O'Leary, Santiago Signorelli, A Harvey Millar (2022) Alternative oxidase (AOX) 1a and 1d limit proline-induced oxidative stress and aid salinity recovery in Arabidopsis, Plant Physiol, 188(3):1521-1536.

Elena S Belykh, Ilya O Velegzhaninov, Elena V Garmash (2022) Responses of genes of DNA repair, alternative oxidase, and pro-/antioxidant state in Arabidopsis thaliana with altered expression of AOX1a to gamma irradiation, Int J Radiat Biol, 98(1):60-68.

Shoya Yamada, Hiroshi Ozaki, Ko Noguchi (2020) The Mitochondrial Respiratory Chain Maintains the Photosynthetic Electron Flow in Arabidopsis thaliana Leaves under High-Light Stress, Plant Cell Physiol, 61(2):283-295.

Xinyan Qiao, Mengjiao Ruan, Tao Yu, Chaiyan Cui, Cuiyun Chen, Yuanzhi Zhu, Fanglin Li, Shengwang Wang, Xiaofan Na, Xiaomin Wang, Yurong Bi (2022) UCP1 and AOX1a contribute to regulation of carbon and nitrogen metabolism and yield in Arabidopsis under low nitrogen stress, Cell Mol Life Sci, 79(1):69.

Dinakar Challabathula, Benedict Analin, Akhil Mohanan, Kavya Bakka (2022) Differential modulation of photosynthesis, ROS and antioxidant enzyme activities in stress-sensitive and -tolerant rice cultivars during salinity and drought upon restriction of COX and AOX pathways of mitochondrial oxidative electron transport, J Plant Physiol, 268:153583.

Adrien Burlacot, Ousmane Dao, Pascaline Auroy, Stephan Cuiné, Yonghua Li-Beisson, Gilles Peltier (2022) Alternative photosynthesis pathways drive the algal CO 2-concentrating mechanism, Nature, 605(7909):366-371.

Jaideep Mathur, Olivia Friesen Kroeker, Mariann Lobbezoo, Neeta Mathur (2022) The ER Is a Common Mediator for the Behavior and Interactions of Other Organelles, Front Plant Sci, 13:846970.

Jie He, Nico Rössner, Minh T T Hoang, Santiago Alejandro, Edgar Peiter (2021) Transport, functions, and interaction of calcium and manganese in plant organellar compartments, Plant Physiol, 187(4):1940-1972.

Cristian Chiñas-Palacios, Carlos Vargas-Salgado. Jesus Aguila-Leon, Elias Hurtado-Pérez (2021) A cascade hybrid PSO feed-forward neural network model of a biomass gasification plant for covering the energy demand in an AC microgrid, Energy Conversion and Management, 232: 113896.

Ullah, K.; Jiang, Q.; Geng, G.; Rahim, S.; Khan, R.A. (2022) Optimal Power Sharing in Microgrids Using the Artificial Bee Colony Algorithm. Energies, 15, 1067.

Srilatha Pulipati, Ramasubbareddy Somula, Balakesava Reddy Parvathala (2021) Nature inspired link prediction and community detection algorithms for social networks: a survey, Int J Syst Assur Eng Manag, https://doi.org/10.1007/s13198-021-01125-8.

6. Results
Results need clearly.
-Need nice story of results.
Thanks for your valuable suggestion. We have modified the paper so to better put in evidence the outcome of this review paper.

The most relevant result of this review paper is surveying mechanisms regulating plant cells homeostasis and suggesting the possibility for using them in future studies on microgrids.

7. Other
Scientific Soundness: Low.

Flexibility of energy homeostasis in plant cells is suitable for adaptation to fluctuating environment and such characteristics might be important for design of future microgrids. Mechanisms underlying this flexibility of plant cellular systems are revealed by many studies. Thus, surveying mechanisms regulating plant cells homeostasis by this review will provide useful information to inspire new idea to design microgrids.
Also, the main concept of this review paper is mentioned in the line 601-606 in the conclusions.

8. English
-English language and style are fine/minor spell check required.

Thanks for appreciation. A general check of English has been done to improve spelling

9. Conclusion
-You need to write more of the conclusion part.
-Future work" write more. (Must be improved)
-Conclusions supported by the results. (Write more)

Thanks to your valuable suggestion, the conclusions have been extended and improved, putting in evidence the cornerstones of the review work and the potential for future developments. The usefulness of the proposed review as a reference work surveying mechanisms regulating plant cells homeostasis and offering a new point of view on future research on microgrid is addressed as well, as suggested by the Reviewer.

Round 2

Reviewer 3 Report

Dear Authors.

The revision adequately address the concerns expressed in last review. So, I recommend that this revised manuscript can now be recommended for publication (Accept as is).

This manuscript is a resubmission of an earlier submission. The following is a list of the peer review reports and author responses from that submission.

Round 1

Reviewer 1 Report

The study of this review paper is well organized and it could be attracted more readers. The present form of a manuscript can be accepted for publication.

Author Response

Thank you very much for your positive comments, and thank you very much for your suggestion. It is interesting to compare the efficiency of energy storage systems. However, we still have the gaps in the characteristics between plants and microgrids, and it is still not easy to compare them. Therefore, we will provide further information in future studies.

Reviewer 2 Report

The authors have not really revised the manuscript in the necessary way. Rather than, as suggested, explicitly defining the analogous structures and processes in plant metabolism and energy grids, the authors added a short and insufficient description of grids (e.g. lines 54, 118) that fails to clearly outline the necessary parallelisms. There is no recognition the authors have seriously reevaluated their “framework” in light of the extensive previous comments on useful methods to examine bioinspired problems, or acquired the necessary knowledge to properly define microgrids. As a result, there is no way to clearly determine the principles from plants that can be translated to microgrids, their utility, and potential caveats and limitations. In addition, the lack of a more complete description of microgrids means the specific problems or inadequacies of current technology is not addressed, and so the improvement that can be gained from a new bioinspired design approach is unclear.  As in the previous ms, the authors descriptions of plant metabolism concentrate on components, and ignore mechanisms that would, if transferred to grids, provide for important functions. The necessity of, and lack of, complete structure-mechanism-function descriptions was noted in the previous version and the authors have appeared not to have consulted any literature referenced that might enable them to move forward here.

I have now furnished a series of very specific questions for the authors that arise at particular points in the ms. Progress towards an acceptable ms will occur only if the authors can satisfactorily respond to these concerns, all of which relate generally to the issues noted above.

  • Re: Fig 1: this figure lacks a description. What do the arrows mean? What is being communicated and which components are providing or processing information? Why are there three microgrids and what is the relationship between them? The lack of description means this figure tkes up space but contributes little to understanding.
  • L 109: Despite making this comment previously, it is not clear what is the analogy between plants and microgrids. Are the energy producing/storage entities in a plant equivalent to an energy producing/storage unit in a microgrid, or the microgrid itself? Are principles from plants being used to facilitate coordination within a microgrid, between different microgrids, or both? None the subsequent discussion identifies the level of communication (within vs. across). What are the differences between microgrids and plants that make applying communication/feedback in plants problematic, and in particular, how relevant is it that in plants, all energy/producing units all work for the benefit of the plant whereas these components all are working to "benefit" independent entities that may not share the same goal? How does this restrict or change the applicability of principles? These questions must be addressed clearly and completely for the ms to have utility, and many are repeated in specific sections below.
  • L156: Specifically how? What is being regulated? What parameters are providing feedback? Why is the current method of doing this insufficient and what is it it about mechanisms of regulation in plants that suggests application of these principles will help address the problem?
  • L173: why would this (negative feedback) be useful in microgrids and an advancement over current approaches? What parameters would be monitored and what processes would be affected?
  • L185: Related to the above, it is not clear why current methods of regulation in smart grids are problematic, nor is it clear that principles from plants are better.
  • Line 192: what does “finely tuned” mean? What property is being measured to make this statement and what are the consequence? Do plants have more rapid responses, are they more sensitive to environmental conditions, are responses more accurate?
  • L236: It’s not clear whether the authors are suggesting the same type of sensors would be useful, or that the information obtained can be used in a beneficial way. If the former-how is this possible given differences in plant vs. human technological sensing? If the latter, what parameters would be monitored, why, what are the benefits, and how is this an improvement on existing systems?
  • L247-252 What parameters are providing feedback to allow this to happen and how would this be translated to microgrids? What components in a microgrid would perform an equivalent role to TOR, what signals would be sensed and what information would provide feedback? What is the target of regulation and what is the goal function?
  • L256: What mode would this be and why would this be advantageous? How is this useful when end users can do this themselves by regulating energy consumption themselves? Evidence needs to be furnished that microgrids currently do not do this. For instance, roof top solar systems provide energy when the sun is shining, and either send this to the residence for use and storage or to the grid if use/storage is not required. How is this different from what plants do?
  • Line 261: See point #8
  • L 265: See point #8
  • L267: So is the suggestion that the proposed translation to microgrids involve monitoring of different sorts of signals? What might these be, what information is being used, how do different sources of information interact, and what is the goal function? Why is this a useful advancement on current methods?
  • L281: What is being modeled here, i.e. what parameters provide indication of better performance when applied to microgrids and what sorts of drivers/inputs are being altered and how does this apply to microgrids?
  • L304: I see no connection to grids. How does understanding how plants heal inform microgrids when the structural components are so different? What are the equivalent/analogous structures and processes in microgrids for what is being described? How does this fit the definition of "healing" which is to regenerate structures so they can regain function? What would be the equivalent signals corresponding to damage or degradation?
  • L340: Is the claim here that microgrids lack storage? This seems hard to accept and is directly contradicted by Fig 1.
  • L345: Why not simply have a backup generator then? There is a lot of conceptual casualness in this section that makes it very un-useful.
  • L 353: This statement lacks any sort of support. How is it even possible to assess the feasibility here without identifying specific components of the grid and the energy flows? It's not clear that there is enough waste heat to recover, or those systems (even theoretical ones) that could do this. The level of vagueness makes this statement completely without utility.
  • L384: This is one of the few times the text provides sufficient details of microgrids. However, what's missing here is the analogous processes in plants and why they might result in better regulation. What are the differences in hierarchical control between plants and grids (notably-the text does not even establish plants have hierarchical control)? What are the corresponding feedback mechanisms, targets of feedback, signals used to provide feedback, goal or performance functions, roles of positive vs negative feedback etc...
  • L402: The model lacks utility until the previous questions have been answered.
  • L443: Why would this in anyway be helpful when the structural components of microgrids do not involve chemical kinetics? What is the general process that is important here and how will knowing the specifics in plants allow for translation of principles to microgrids? What would these principles potentially be?
  • L470: This is vague as it does not firmly suggest analogous processes and structures in plants and microgrids, does not identify how to translate model results when the time courses, types of information being monitored and performance measures are all different. There is no utility to the approach in light of the absence of these parallels. Problem and solution driven approaches are not defined and their potential contributions to this analysis unclear.  Simply mentioning them adds nothing.

Author Response

Thank you very much for your valuable comments. Although the scope of this review is not to propose a definite bio-inspired model, it can be the initial step to suggest and spread a new idea. In other words, the scope of this paper is to catch the possibilities offered by the plant cell energy mechanisms to be exploited for defining behavioral models or control/management strategies for electrical microgrid. Therefore, we would like to publish it as open questions for the research communities to trigger the discussion. Unfortunately, the research has not yet reached a sufficient level of maturity to allow us to answer questions raised by you. But, most of questions raised may be addressed by the research community by opening the ideas that are described in this review.

The point is if such a kind of contribution can be considered acceptable by the journal. If so, we would ask you to reconsider your evaluation on the basis of the intended scope of the paper.